# Spatiotemporal Pattern and Driving Factors of Urban Sprawl in China

**Xin Zhang**  **and Jinghu Pan** * 

College of Geography and Environmental Science, Northwest Normal University, Lanzhou 730070, China; 2019212415@nwnu.edu.cn
* Correspondence: panjh_nwnu@nwnu.edu.cn

**Abstract:** Urban sprawl is a complex phenomenon related to abnormal urbanization, and it has become a key issue of global concern. This study aimed to measure urban sprawl in China and explore its spatiotemporal patterns and driving factors. Based on 343 Chinese cities at the prefecture level and above, remote sensing-derived data from 2000 to 2017 were used to calculate the urban sprawl index (USI). The evolutionary trend and spatiotemporal pattern of urban sprawl in China were then analyzed using trend analysis and exploratory spatiotemporal data analysis, and Geodetector was applied to investigate the factors driving the changes. The results show the following. ① Moderate or high urban sprawl development occurred in China from 2000 to 2017. In terms of spatial distribution, the USI was high in northwest China and low in southeast China. ② The local spatial stability of the USI gradually decreased from southeast to northwest and northeast. USI had strong spatial dependence. No significant spatiotemporal transitions in urban sprawl were observed, and the spatial pattern was stable with strong spatial cohesion. ③ The gross regional product (GRP) of the tertiary industry, the total GRP, and investment in real estate development have been the most important factors affecting sprawl in cities at the prefecture level and above in China.

**Keywords:** urban sprawl; spatiotemporal pattern; driving factor; Geodetector; China

## 1. Introduction

Urban sprawl, a trend of abnormal and unrestricted urbanization development, has been a controversial topic [1], and it continues to present a challenge to countries across the world. Urban sprawl refers to unsustainable spatial expansion in a city in the process of development: it tends to be random and unplanned, scattered and discontinuous, associated with strong dependence on transportation for travel, and characterized by a single land-use type and severe land-use conflicts [2]. With the advancement of industrialization, the improvement of highway transport facilities, and the increased use of motor vehicles, suburban areas have rapidly developed, urban centers have tended to decline, and sprawl has gradually become a prevalent urban development issue in developed Western countries [3]. Since the initiation of the "Reform and Opening-Up" policy, urbanization has advanced extremely rapidly in China [4]. In 1978, the urbanization rate, calculated as the resident population, was only 18%, and by 2019, it had increased to 60.6%, which requires a larger urban capacity to accommodate the dramatically increasing urban industry and population. In 2019, the per capita construction land in Chinese cities was as high as 200 m², which was much higher than that in developed countries. The outward expansion of the city has become the "normal" condition of many cities, and the blind expansion of built-up areas is ubiquitous [5]. Urban sprawl usually has adverse effects on the development of a city, causing issues such as traffic congestion, environmental pollution, and social differentiation [6–8]. In addition, the outward expansion of a city often occurs at the expense of the quantity and quality of cultivated land. The increase in the city scale also increases the cost of public infrastructure construction, and inadequate infrastructure may lead to the "empty city" or "ghost city" phenomenon [9,10], reducing the land-use

efficiency. Furthermore, urban sprawl also significantly affects the ability to realize sustainable development. For example, insufficient ecosystem services related to phenomena of uncontrolled urban expansion in cities have prevented many countries from making significant progress toward sustainable development goals [11]. The economic impacts of urban land-use transformation are complex and debated [12], and urban expansion and socioeconomic transformations may negatively impact the environmental quality and functions of peri-urban landscapes [13]. The continuous emergence of the above drawbacks has attracted the attention of the government, the public, and scholars [14].

Many scholars have carried out in-depth research on issues related to urban sprawl, largely focusing on its definition, evaluation methods, internal mechanism, regulatory measures, and ecological and environmental effects. Defining urban sprawl is the primary task of urban sprawl research [15]. Although scholars have an in-depth understanding of urban sprawl, due to its complexity and multidimensional nature, a unified definition of urban sprawl has not yet been established. At present, related concepts are mainly defined on the basis of the manifestations, formation characteristics, and impacts of urban sprawl [16]. In the existing literature, the sprawl in Western countries is generally described as the expansion of cities beyond urban boundaries to extra-urban areas, accompanied by low density and transport dependency [15,17]. However, because developing countries are in a stage of rapid urbanization, they differ from Western countries in terms of population density, urban–rural structure, land system, and socioeconomic development level. China is no exception. Compared to the definition of sprawl in Western countries, the definition in China is more specific and focuses on describing the manifestations of a certain type of urban sprawl. Yue et al. [18] defined urban sprawl as a low-density type of urban expansion occurring beyond the urban built-up area, including low-density edge-growth or leapfrog growth, such as industrial development zones or college towns. However, many scholars confuse urban sprawl with urban expansion; this paper distinguishes between these terms. Urban expansion characterizes the pursuit of "absolute scale" through spatial–temporal changes in urban land use. In contrast, urban sprawl describes the changing development of cities in terms of three dimensions: low density, scale-up, and negative impacts, which can reflect imbalances and inadequate development. Therefore, it is inaccurate to treat urban expansion and urban sprawl as the same phenomenon.

The lack of defined measures of urban sprawl is a difficult problem. Measures of urban sprawl have diverged as the concept has evolved. In particular, China and the West differ in their concept of urban sprawl and, hence, in its measures. Western scholars have calculated the urban sprawl index by applying a quantitative metric in combination with space. Early studies used a single-indicator measure, but with increasing understanding of sprawl and the region under study, a multi-indicator approach was gradually adopted to construct a comprehensive urban sprawl index. The study of urban sprawl in China started relatively late, and the measures were more haphazard. Urban sprawl in China is mostly based on single-indicator measures. For example, Lang and Lefurgy [19] measured urban sprawl with population density indicators in the United States. Taiwo [20] constructed the Urban Sprawl Index, which is the ratio of the area growth rate of built-up area to the population growth rate (elastic coefficient), to measure the urban sprawl in Nigeria. However, the above studies failed to distinguish between uniform and concentrated population distributions within a city. Scholars have proposed urban sprawl indices that account for differences in population distribution within cities. Lopez and Hynes [21] and Fallah et al. [22] constructed a sprawl index based on the spatial distribution of population density within a city and subtracted the population proportion of low-density areas within the city from that of high-density areas to reflect the extent of urban sprawl in the United States. Li and Li [8] used the difference between the urban area growth rate and the population growth rate to measure urban sprawl in China. If the proportion of a low population density area is relatively large, then the intensity of land use is low, and the degree of urban sprawl is high. Scholars have since improved the above indicators in an attempt to obtain urban sprawl measurements that better reflect the spatial details of cities

and are less disturbed by the abnormal distribution of the local population. Based on the analysis of urban land sprawl and urban population sprawl, Qin et al. [23] constructed a new urban sprawl index to study urban sprawl in China.

After identifying an urban sprawl measurement method, Chinese and Western scholars gradually focused their research efforts on the spatial characteristics presented by urban sprawl. In general, spatial characteristics are relatively easy to observe, such as low-density development, leapfrog or scattered development, and poor accessibility [16,18]. Later, scholars added a temporal dimension on this basis and explored the spatiotemporal evolution characteristics of urban sprawl, which has strengthened our research to some extent. Nazarnia et al. [24] explored the spatial pattern of urban sprawl in Montreal, Quebec City, and Zurich from 1951 to 2011. Feng et al. [3] investigated the spatial correlations and spatial distribution patterns of urban sprawl in China by utilizing Global Moran's I (GMI) and Local Moran's I (LMI). The study of spatiotemporal changes has been relatively generalized, and remains superficial, and the study of the spatiotemporal pattern of urban sprawl has received little attention.

As research has become more in-depth, understanding the intrinsic mechanism of urban sprawl has also become a research hotspot. In the long term, identifying the drivers of urban sprawl is important to promote urban development policies. According to the existing literature, the driving forces of urban sprawl can be explored from political, economic, sociological, and environmental aspects [3,8,25]. In Western countries, market forces, consumer preferences, public subsidies, and land-use regulations are considered the main drivers of urban sprawl [26]. Burchfield et al. [27] reported that ground water availability, temperate climate, rugged terrain, decentralized employment, early public transport infrastructure, uncertainty about metropolitan growth, and unincorporated land in the urban fringe all increase sprawl. Research by Pirotte and Madre [28] showed that the French urban development model is highly dependent on the density of surrounding farms and their ability to provide facilities. Because of its very different institutional context, the causes of urban sprawl in China include many aspects, among which government regulatory factors, economic factors, and social factors predominate [3,29]. Most of the related research has been qualitative, and quantitative and spatialized studies at large scales have been extremely rare. Yue et al. [24] discussed the impacts of farmland preservation policy, population policy, and urban planning on urban sprawl in Hangzhou City. Overall, the research on urban sprawl has mostly focused on a single city (or urban agglomeration) at a small scale [30–32], whereas analysis of spatial–temporal differentiation and spatial–temporal correlation characteristics of urban sprawl at a large scale has been overlooked.

Based on the above analysis, the urban sprawl indices of 343 Chinese cities at the prefecture level and above were analyzed. This study included three main parts. First, we established an urban sprawl index capable of capturing the detailed differences that define inner-city areas by using nighttime light remote sensing data and LandScan population spatial distribution data. The index was further used to measure urban sprawl in China between 2000 and 2017. Second, we analyzed the spatial–temporal patterns by applying the Exploratory Spatial–Temporal Data Analysis (ESTDA) method. Third, we explored the drivers of urban sprawl. The ultimate goal of the third part was to provide decision-making references for New Urbanization Construction and Regional Planning.

## 2. Data Sources

### 2.1. Data

Four types of data were used in this analysis: Nighttime Light, LandScan Global Population, Administrative Boundary, and Statistical Data. Nighttime Light Data were collected from the National Oceanic and Atmospheric Administration/National Geophysical Data Center (https://www.ngdc.noaa.gov/eog/download.html (accessed on 20 March 2020)). The annual cloud-free-composited stable NTL imagery obtained by DMSP-OLS is $30 \times 30$ arc-seconds gridded cell-based nocturnal luminosity spanning from 2000 to 2013 with DNs ranging from 0 to 63. The VIIRS products contain spatially gridded nocturnal

radiance values at a spatial resolution of 15 arc-seconds across the latitudinal zone of 65° S–75° N. The monthly "VIIRS Cloud Mask (vcm)" version was used, which excludes observations affected by stray light, with a time series of 2012–2017. LandScan Global Population Data were collected from Oak Ridge National Laboratory's website (available at https://landscan.ornl.gov/landscan-datasets (accessed on 27 March 2020)). The above three sets of data are grid data. Administrative Boundary Data were collected from the National Geomatics Center of China (available at http://www.ngcc.cn/ngcc/ (accessed on 20 March 2020)) and are vector data. All of the driving forces analyzed in this study are from China City Statistical Yearbook, which is statistical data. The evaluation units were 343 cities at the prefecture level and above in China.

*2.2. Data Processing*

DMSP-OLS NTL data have disadvantages, such as their tendency for saturation and the discontinuity of multiplatform data [33]; as a result, these data need to be preprocessed. Projection conversion, pixel resampling, area clipping, desaturation spillover effect, and interannual continuity corrections were successively conducted to obtain data that meet the research requirements. As a result, corrected DMSP-OLS data were obtained for 2000–2013 [34]. Then, the NPP-VIIRS monthly NTL data were converted to projection coordinates, resampled, and clipped. The annual mean data of NPP-VIIRS NTL from 2013 to 2017 were obtained by removing background noise and abnormal pixels, correcting relative radiation, and eliminating unstable light sources.

Since DMSP-OLS NTL data were not released after 2013, and NPP-VIIRS NTL data have only been provided since 2012, in order to study a longer time series, it was necessary to integrate the two data resources [35]. Referring to the data integration method of Li et al. [36], a fitting equation was established for the two datasets for coinciding years using a power function, and then the image was denoised by a Gaussian low-pass function [37]. The 2013 DMSP-OLS NTL image was used as the reference, and the NPP-VIIRS NTL data of 2014–2017 were corrected using the fitting equation, resulting in the simulated DMSP-OLS NTL image with 1 km spatial resolution for 2014–2017. Finally, nighttime light data with a spatial resolution of 1 km in China from 2000 to 2017 were obtained.

The extraction of urban boundaries is an important step in the study of urban sprawl. With the continuous enrichment of NTL products, the use of NTL data to extract urban areas has attracted widespread attention from scholars who study urban sprawl [18,38]. In order to exclude city areas that are developed but lack population activities, the developed areas in the suburbs were also included. In this paper, areas meeting both the nighttime light coverage and population distribution criteria are defined as urban areas. Because of their urban geographical locations and economic development, Beijing, Shanghai, Guangzhou, Shenzhen, and Chongqing were used as the samples. MATLAB software was used to determine the best threshold, and the average value of the threshold of each city was taken as the best threshold of the national urban area in that year [39]. Fifteen cities were used for verification, and the city areas extracted by the optimal threshold in 2017 were compared with those extracted by visual interpretation of high-resolution images. The overall accuracy is above 83.24%, which meets the accuracy requirement of this study. Secondly, referring to the standard proposed by Mao et al. [40], a population density of 1000 person·km$^2$ was used as the threshold of urban regional division. Finally, the two sets of data were superimposed to obtain a more accurate urban boundary of each city.

## 3. Methods

The method involved three steps: (1) quantitative measurement of urban sprawl in China after data processing; (2) exploration on the spatial-temporal patterns of urban sprawl using the ESTDA method; (3) estimation of the impacts of economic, social and government regulation factors on urban sprawl using Geodetector model. For a more intuitive presentation of the workflow, a flow chart diagram is shown in Figure 1.

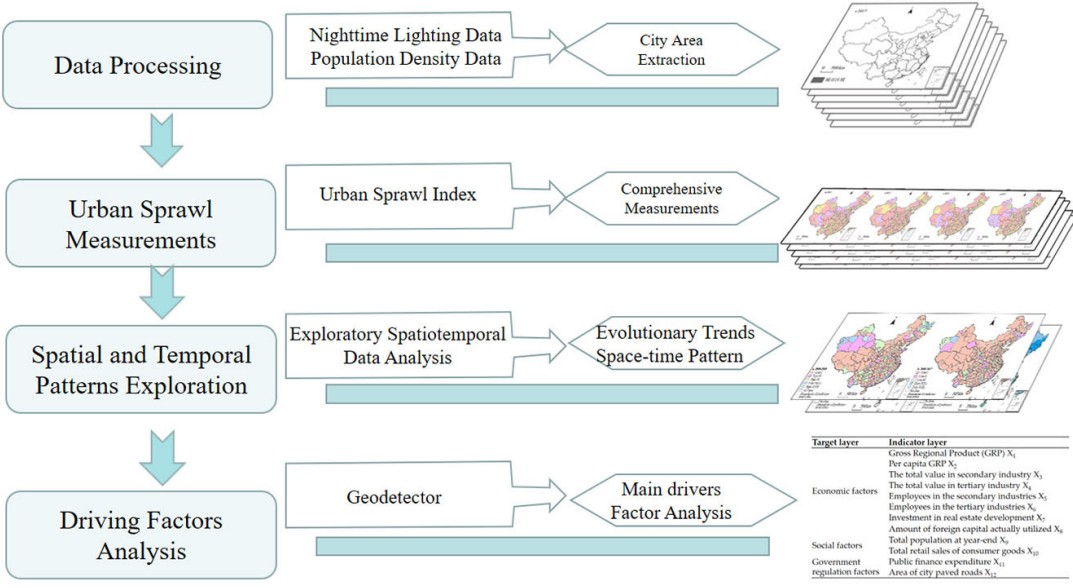

**Figure 1.** Flow process chart.

### 3.1. Urban Sprawl Index

Combined with the low-density expansion of urban sprawl, both urban population and land area aspects were used as a basis for constructing the sprawl index. If the proportion of the population in low-density areas of a city increases, or if the living area proportion of the population in low-density areas increases (which means that the density of urban low-density areas continues to decrease), then the degree of urban sprawl is rising [23]. The formula of the urban sprawl index (USI) is

$$USI_i = \sqrt{USA_i \times USP_i} \tag{1}$$

$$USA_i = 0.5 \times (LA_i - HA_i) + 0.5 \tag{2}$$

$$USP_i = 0.5 \times (LP_i - HP_i) + 0.5 \tag{3}$$

where $USA_i$ and $USP_i$ denote area sprawl and population sprawl, respectively. $LA_i$ is the ratio of the area of low population density in city i to the total area occupied by urban land in the city. $LP_i$ is the population ratio between the low-population-density area in city i and the city as a whole. $HA_i$ is the ratio of the area of high population density in city i to the total area occupied by urban land in the city. $HP_i$ is the population ratio between the high-population-density area in city i and the city as a whole. A low-population-density area in a city is defined by a population density below the national average, whereas an area with high population density is above the national average. The urban sprawl index ranges from 0 to 1, and the closer the value is to 1, the more intense the urban sprawl is.

In contrast to the commonly used average density method to measure urban sprawl, the USI captures detailed differences within urban regions. It does not regard the city as a homogeneous area, thus preventing abnormally high or low densities in the city from influencing the overall density of the city; thus, the USI is more reliable for estimating the urban sprawl.

### 3.2. Exploratory Spatiotemporal Data Analysis (ESTDA)

In this study, the spatial pattern differentiation and temporal evolution characteristics of urban sprawl at the global and local levels were comprehensively analyzed through ESTDA. The ESTDA method effectively integrates time and space and realizes space–time interaction analysis [41]. Local Indicators of Spatial Association (LISA) time path analysis was used to study the dynamic migration law of local autocorrelation in a Moran scatter

diagram from the perspective of time evolution by introducing the time dimension. In addition, with the use of the paired migration analysis of the attribute values and the spatial lag values of the USI of each city in the time series, changes in the spatial–temporal interaction and the dynamic characteristics of spatial–temporal differences in urban sprawl at the local level were explored and explained. Thus, the continuous expression of the local spatial dependence from the "instantaneous scene" to the "interactive dynamic scene" was realized. The LISA time path is generally described by relative length and tortuosity. Rey et al. [42] improved the classical Markov chain by including such factors as the migration path, orientation, and aggregation of each measurement unit in a Moran's I scatter diagram, and they defined the concepts of local Markov migration and space–time transition. The space–time transition is divided into four types: I, II, III, and IV. Rey and Janikas [41] proposed spatiotemporal flow and spatiotemporal aggregation to characterize the relationship between the number of specific transition types in the study period and the number of all transitions in the global range. Further details are provided in the Methods section.

### 3.3. Indicator System of Driving Factors

Urban sprawl and its regional differences are affected by many factors. Based on previous studies [3,4,8,18,43,44], 40 candidate factors affecting urban sprawl from three aspects—economy, society, and government regulation—were selected. Then, the canonical correlation coefficient between each candidate factor and the dependent variable was calculated by performing canonical correlation analysis, and the factors with a small correlation coefficient were excluded. An index system of 12 influential factors with a high correlation with the USI was finally obtained (Table 1).

**Table 1.** The driving factors of USI.

| Target Layer | Indicator Layer | Unit |
|:---:|:---:|:---:|
| Economic factors | Gross Regional Product (GRP) $X_1$ | CNY 10,000 |
| | Per capita GRP $X_2$ | yuan |
| | The total value in secondary industry $X_3$ | CNY 10,000 |
| | The total value in tertiary industry $X_4$ | CNY 10,000 |
| | Employees in the secondary industries $X_5$ | person |
| | Employees in the tertiary industries $X_6$ | person |
| | Investment in real estate development $X_7$ | CNY 10,000 |
| | Amount of foreign capital actually utilized $X_8$ | USD million |
| Social factors | Total population at year-end $X_9$ | 10,000 persons |
| | Total retail sales of consumer goods $X_{10}$ | CNY 10,000 |
| Government regulation factors | Public finance expenditure $X_{11}$ | CNY 10,000 |
| | Area of city paved roads $X_{12}$ | $hm^2$ |

### 3.4. Geodetector

Geodetector is a statistical method for detecting spatial heterogeneity and the degree of influence of driving factors on dependent factors [43]. Factor detection was used in this study to identify factors that affect the spatial–temporal differentiation pattern of urban sprawl, and interaction detection was used to explore the degree of influence of interactions among factors on the urban sprawl. Factor detection uses the q value to describe the extent to which a certain factor X explains the spatial differentiation of attribute Y. Interaction detection was used to measure the interaction between factors.

## 4. Results

### 4.1. General Characteristics of Urban Sprawl

In this study, 343 Chinese cities at the prefecture level or above were the research object. The USI of each city in the country was calculated for the period from 2000 to 2017 using Formula (1), and the spatial distribution of the USI in each year was obtained. Combined with the data distribution, to facilitate comparison, the natural discontinuity

method and manual classification were used to divide the USI into five categories based on data from 2008, and the classification standard was applied to all other years between 2000 and 2017 to compare the spatial and temporal patterns of urban sprawl in China (Figure 2). Overall, the national average USI was declining over the 18 study years, from 0.483 in 2000 to 0.468 in 2017. The average value of the USI between 2000 and 2017 was 0.476, which places it in the middle class and close to the fourth class of the urban sprawl index classification, indicating that Chinese city sprawl was above the intermediate level. The geographical differences in urban sprawl were significant; the extent of urban sprawl in the central and western regions was greater than that in the eastern and northeastern regions.

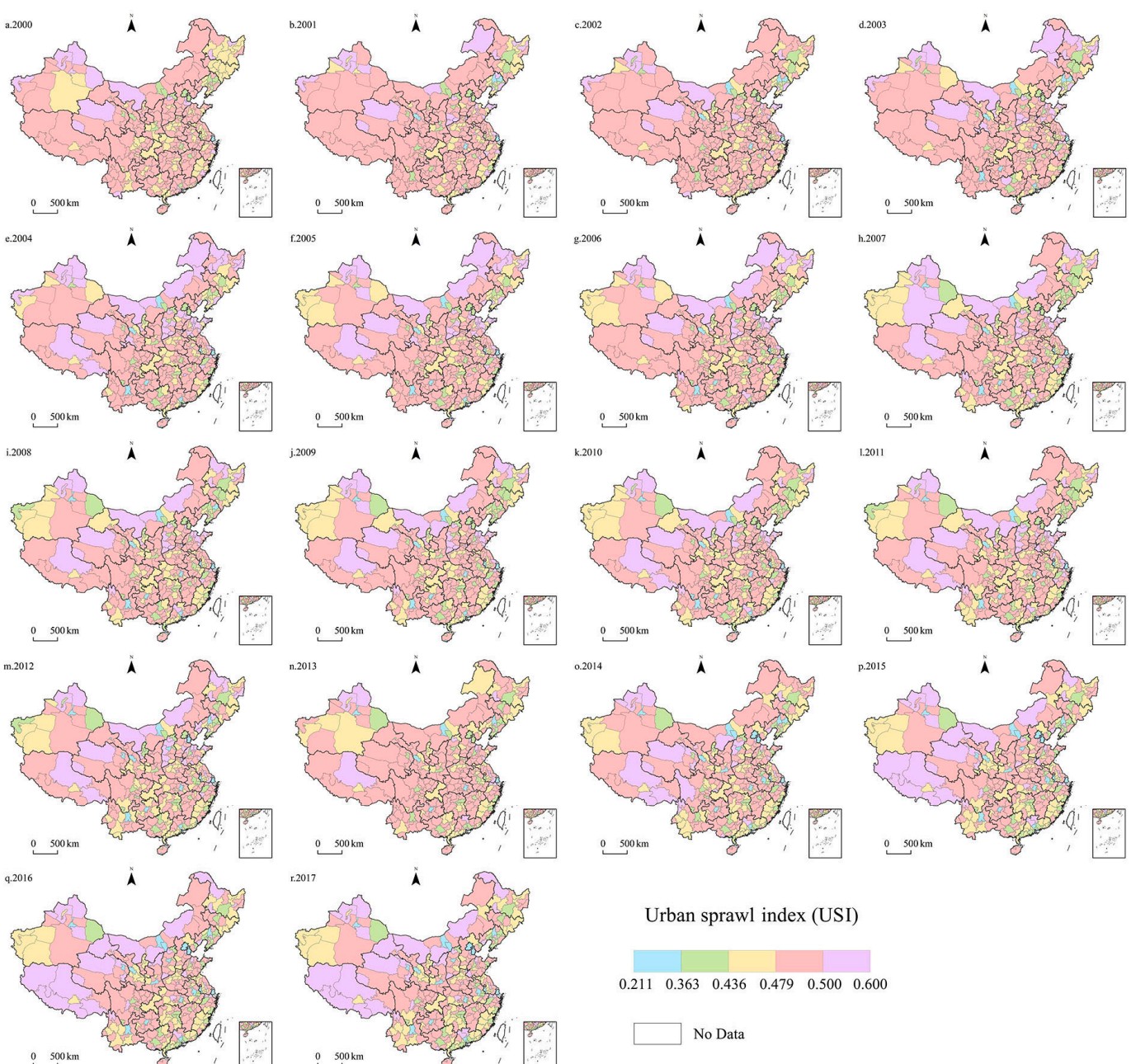

**Figure 2.** Spatial distribution of USI in China from 2000 to 2017.

The changing trend of urban sprawl (Figure 3) was obtained by calculating the interannual rate of sprawl from 2000 to 2017. To present results that are clearer and more intuitive, the quantile method was used to divide the data into four types: mild reduction, essentially

unchanged, mild growth, and significant growth. Figure 3 shows that the USI of most prefecture-level cities (162) in China remained largely unchanged, with 85 cities exhibiting a significant growth trend. Of the top 20 cities with the greatest increase in sprawl, 9 were in the western region and 7 were in the central region. Cities with significant growth were mostly distributed near interprovincial borders, such as the Hunan–Guangdong–Jiangxi, Anhui–Henan–Shandong, Zhejiang–Anhui–Jiangsu, Guizhou–Sichuan, and Hebei–Shandong borders. Regions around interprovincial borders are frequently neglected in regional economic development, and they show a lag and marginalization pattern compared with more developed regions within a province. This study shows that this region also tended to have the most drastic sprawl of cities. Mild reduction in the degree of sprawl was observed in a small number of cities, particularly provincial capital cities, or regional central cities.

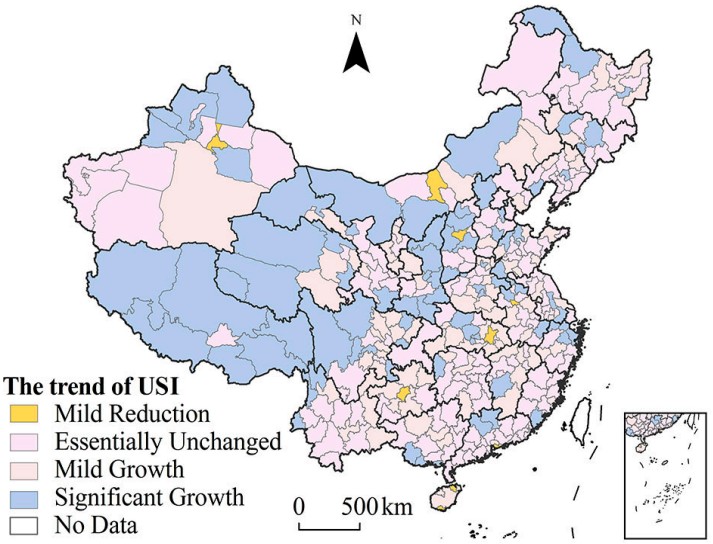

**Figure 3.** Spatial distribution of USI in China from 2000 to 2017.

### 4.2. Spatial and Temporal Patterns of USI

The spatial autocorrelation of the USI from 2000 to 2017 was measured by using the Moran's I index. The global Moran's I index and z value were greater than 0 in each year, and the *p*-value reached a confidence level of 99.9%, which indicates that the USI had high spatial autocorrelation and spatial dependence in the 18 study years. The relative length of the LISA time path and the curvature of the USI in various cities were calculated and classified into five types using the natural interrupted point method: low, relatively low, medium, relatively high, and high.

As observed in Figure 4a, from 2000 to 2017, the relative length of the USI showed an overall increasing trend from southeast to northwest and northeast, with a higher value in the north than in the south and a higher value along the coast than inland. This indicates that the urban sprawl in the southeastern region had a more stable local spatial structure. Eight cities have paths categorized as relatively high, and they are grouped into two categories: Karamay, Shihezi, Jiayuguan, and Fushun are important resource-based cities for oil and steel in the region; the city scale is generally not large, but the development is rapid. Dongguan, Shenzhen, Zhuhai, and Sanya are all coastal open cities with considerable geographical advantages.

As shown in Figure 4b, from 2000 to 2017, the curvatures of the USI in 343 cities were all greater than 1, which indicates the strong spatial dependence of urban sprawl. The number of cities with local indicators of spatial association curves that moved from low to high present a pyramid structure, and the cities with a low curvature account for 79.3% of the total number, indicating that the USI in most cities showed relatively weak volatility [16]. Cities with a relatively high degree of curvature were mainly concentrated

in the provincial border areas and Hainan province. These cities were subject to the spatial spillover or spatial polarization of neighboring cities and had high volatility in the spatial dependence direction. In other words, they are characterized by relatively high volatility in the process of urban sprawl.

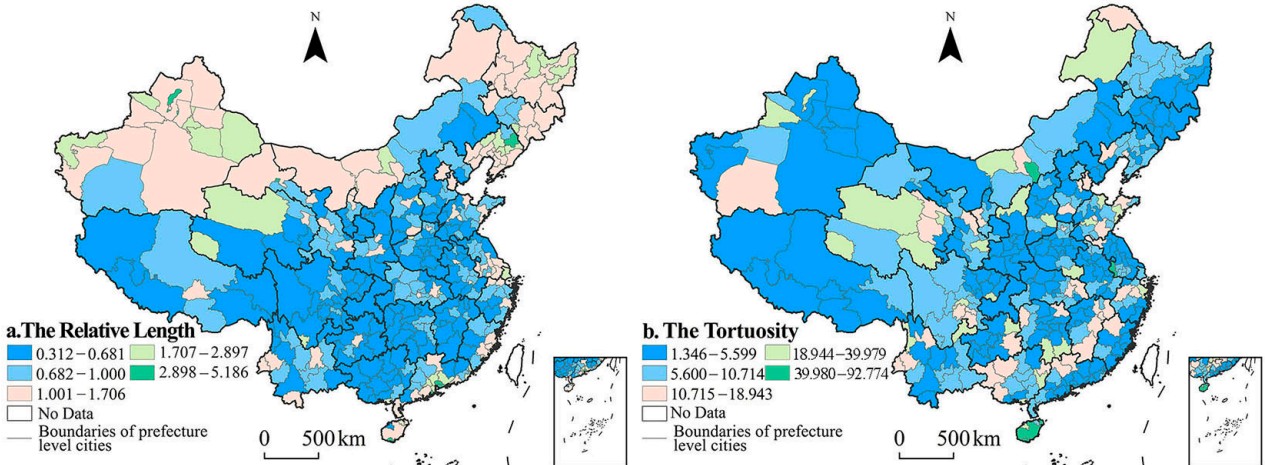

**Figure 4.** LISA time path of USI from 2000 to 2017 in China.

The results of space–time transition are shown in Table 2 and Figure 5. There was no significant spatial–temporal transition in urban sprawl from 2000 to 2008 or from 2008 to 2017. Moran's I scatter is always located in the same quadrant, which is type I transition. The probability of urban sprawl in the two periods was 64.4% and 74.6%, respectively, which indicates that the spatial agglomeration of urban sprawl in China was strong, and the amount of spatial migration was small. Moreover, there was a degree of transfer inertia and high stability among different types. If the spatial effect is considered, the relative mobility (0.390) from 2000 to 2008 was greater than that (0.298) from 2008 to 2017. This indicates that the dynamic change in spatial structure decreased and the stability increased in the latter period. Few cities are characterized by type IV transition, which makes it difficult for cities and neighboring cities to change their urban sprawl status at the same time. It is difficult for each city to change its current relative urban sprawl status.

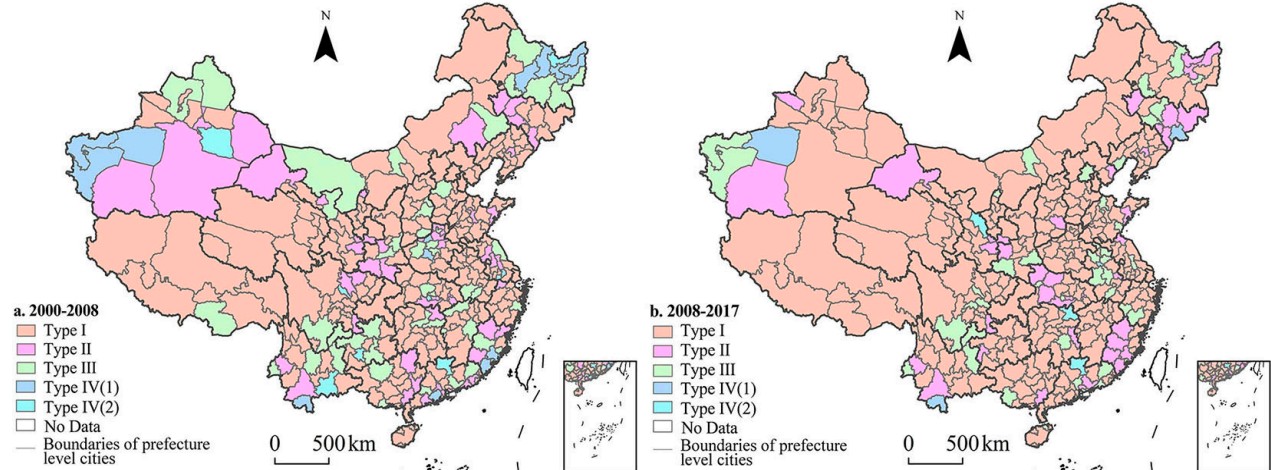

**Figure 5.** LISA Spatiotemporal transition of USI in China.

**Table 2.** Spatiotemporal transition matrices of USI in China.

| Time Interval | 1/t2 | HH | LH | LL | HL | Transition Type | n | Proportion | SF | SC |
|---|---|---|---|---|---|---|---|---|---|---|
| 2000–2008 | HH | 0.730 | 0.085 | 0.053 | 0.132 | Type I | 221 | 0.644 | 0.289 | 0.190 |
| | LH | 0.243 | 0.615 | 0.071 | 0.071 | Type II | 50 | 0.146 | | |
| | LL | 0.137 | 0.118 | 0.510 | 0.235 | Type III | 49 | 0.143 | | |
| | HL | 0.257 | 0.043 | 0.114 | 0.586 | Type IV | 23 | 0.067 | t = 0.390 | |
| 2008–2017 | HH | 0.850 | 0.059 | 0.006 | 0.085 | Type I | 256 | 0.746 | 0.233 | 0.125 |
| | LH | 0.2 | 0.646 | 0.108 | 0.046 | Type II | 39 | 0.114 | | |
| | LL | 0.064 | 0.128 | 0.595 | 0.216 | Type III | 41 | 0.120 | | |
| | HL | 0.192 | 0 | 0.090 | 0.718 | Type IV | 7 | 0.020 | t = 0.298 | |

*4.3. Driving Factors of Urban Sprawl*

The average data for 2000, 2008, and 2017 were used as samples, and Geodetector was utilized to detect the influence degree of individual driving factors and the interaction effects between factors on urban sprawl in China.

4.3.1. Factor Detection Analysis

The factor detection result in Figure 6 shows that economic factors had the greatest impact on urban sprawl. The q value range of each driving factor was 0.227–0.536. The impact on urban sprawl in descending order of strength is as follows: the total value in tertiary industry ($X_4$) > Gross regional product ($X_1$) > Investment in real estate development ($X_7$) > The total value in tertiary industry ($X_3$) > Employees in the tertiary industries ($X_6$) > Area of city paved roads ($X_{12}$) > Total retail sales of consumer goods ($X_{10}$) > Public finance expenditure ($X_{11}$) > Employees in the secondary industries ($X_5$) > Amount of foreign capital actually utilized ($X_8$) > Per capita GRP ($X_2$) > Total Population at year-end ($X_9$).

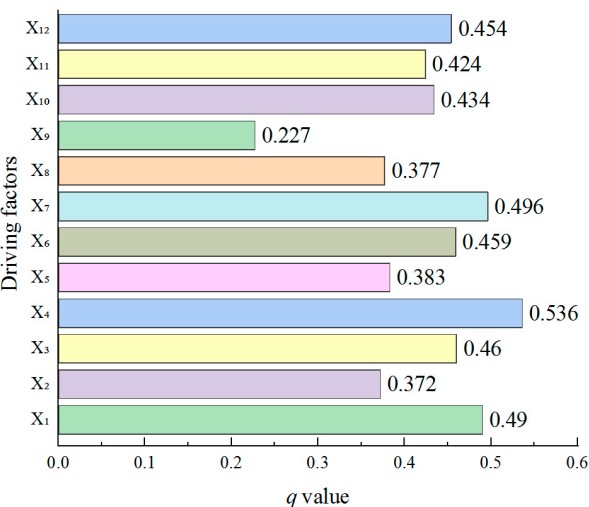

**Figure 6.** Q statistics for factor probes.

The primary driving factor of urban sprawl was the total value of the tertiary industry, with q = 0.536. Actively promoting the tertiary industry is essential for promoting economic development, providing employment opportunities, and increasing residents' income. The cultivation and expansion of professional markets will also help to develop the tertiary industries of transportation, post and telecommunications, financial services, and various intermediary services. These industries have higher requirements for the agglomeration of human resources and capital, which can promote the continuous expansion of the city. The expansion of the city increases demand, resulting in a positive feedback cycle.

The impact of GRP on urban sprawl ranked second among the contributing factors, with a q value of 0.490. GRP has always been considered the most important indicator of

regional economic capacity, representing the level of economic development of a region. The rapid development of the economy may mean that the current development scale is insufficient to meet development needs, and its expansion is promoted in an attempt to create greater benefits. This has stimulated a large increase in the size, volume, and rate of growth of urban land under construction, so the GRP has largely driven the disordered growth of cities. In addition, some local governments blindly consume land resources to elevate their political status, further intensifying urban sprawl.

The third driving factor affecting urban sprawl was the investment in real estate development, with a q value of 0.476. Real estate development investment is closely related to urban sprawl. With the development of urbanization, the population scale continues to grow, and the growing demand for housing necessitates the conversion of substantial amounts of land to residential districts. Land finance has become the main source of government revenue in many cities. Land finance has also been widely used as an important means of urban expansion and development and has become a major promoter of urban sprawl.

### 4.3.2. Interaction Detection Analysis

The interaction detector identifies the strength of the explanatory power of the interaction between each factor on the urban sprawl. As shown in Table 3, the impact of any two factors is the nonlinear growth of urban sprawl. This indicates that each factor has a degree of correlation with urban sprawl, and there is no independent influence factor. The factor interaction is not only greater than the individual effect of a single factor but also greater than the sum of the effects of these two factors. The q values of interactions between factors are more than 0.791, which indicates that the interactions between indicators have high explanatory power on urban sprawl.

**Table 3.** Interaction analysis results.

|  | $X_1$ | $X_2$ | $X_3$ | $X_4$ | $X_5$ | $X_6$ | $X_7$ | $X_8$ | $X_9$ | $X_{10}$ | $X_{11}$ | $X_{12}$ |
|---|---|---|---|---|---|---|---|---|---|---|---|---|
| $X_1$ | 0.490 | | | | | | | | | | | |
| $X_2$ | 0.958 | 0.372 | | | | | | | | | | |
| $X_3$ | 0.791 | 0.961 | 0.460 | | | | | | | | | |
| $X_4$ | 0.829 | 0.957 | 0.886 | 0.536 | | | | | | | | |
| $X_5$ | 0.826 | 0.934 | 0.854 | 0.874 | 0.383 | | | | | | | |
| $X_6$ | 0.926 | 0.983 | 0.968 | 0.942 | 0.925 | 0.459 | | | | | | |
| $X_7$ | 0.902 | 0.872 | 0.883 | 0.883 | 0.879 | 0.932 | 0.476 | | | | | |
| $X_8$ | 0.918 | 0.903 | 0.923 | 0.946 | 0.898 | 0.966 | 0.885 | 0.377 | | | | |
| $X_9$ | 0.961 | 0.976 | 0.964 | 0.964 | 0.952 | 0.938 | 0.974 | 0.965 | 0.227 | | | |
| $X_{10}$ | 0.875 | 0.936 | 0.883 | 0.901 | 0.910 | 0.875 | 0.878 | 0.911 | 0.913 | 0.434 | | |
| $X_{11}$ | 0.918 | 0.903 | 0.938 | 0.937 | 0.858 | 0.900 | 0.906 | 0.903 | 0.936 | 0.855 | 0.424 | |
| $X_{12}$ | 0.857 | 0.964 | 0.824 | 0.866 | 0.930 | 0.933 | 0.852 | 0.940 | 0.932 | 0.899 | 0.885 | 0.454 |

The total population at year-end interacts significantly with other factors, with q values ranging from 0.911 to 0.974. The interactions between the total city population and the gross regional product, the per capita GRP, the total value of the secondary industry, the total value of the tertiary industry, employees in secondary industries, investment in real estate development, and the amount of foreign capital actually utilized were explored. All of these factors explain the urban sprawl with q values greater than 0.9, which indicates that the population index greatly enhances the impact of other indicators on urban sprawl. Population-scale growth requires more resources to meet the needs of residents. This also shows the high development potential of the region. In order to obtain greater development, it is necessary to push the sprawl in the city to the surrounding areas.

The interaction between per capita GRP and other factors is very strong, and the q value range is 0.872–0.983. The amount of foreign capital actually utilized strongly interacts with the other factors, with q values ranging from 0.885 to 0.966. The amount of foreign capital actually utilized mainly acts on the economic and technological development zones,

and economic and technological development zones tend to develop in urban fringe areas. In addition, the injection of foreign capital has attracted a large number of workers and resources, driving economic growth, and has led to the aggravation of urban sprawl.

The total population at year-end, the per capita GDP, and the amount of foreign capital actually utilized has strong interactions with other factors, and the explanatory power of interactions is greater than 87.2%. However, the q values of these factors are all small, which is attributed to the shortcomings of urban sprawl. When a factor acts alone, the impact on urban sprawling is weak, but the explanatory ability is significantly enhanced when it interacts with other factors. These auxiliary factors can enhance the impact of other factors on urban sprawl.

## 5. Discussion

In contrast to the process of urbanization in developed Western countries, which is largely market driven, the driver of urbanization in China is often the government [45,46]. Local governments intervene in the spatial allocation of resources, which in turn determines the formation and development of urban spatial shape through the regulation of the land system and household registration system. Because of the "finance of land" and the pursuit of an expanded scale of urban space [46], in most cities in China, land urbanization increases at a significantly faster pace compared to population urbanization.

### 5.1. Comparison with Other Studies

Because of the complex and multidimensional nature of the urban sprawl problem, most scholars measure it by constructing indicators. Although single-index measures have the benefit of simple calculations, they have a limited ability to comprehensively characterize urban sprawl [47]. The combination of multiple indicators has higher relevance and, in turn, increases the risk of including the causes and consequences of sprawl. In this study, we used indicators of the population and area aspects of the phenomenon of urban sprawl with the aim of reflecting its multidimensional nature. The urban sprawl index described in this paper can subdivide the city based on its internal differences rather than treating the city as a whole, making the measurement of urban sprawl more representative. The results show that geographical differences in urban sprawl are significant. The extent of urban sprawl in the central and western regions was greater than that in the eastern and northeastern regions, which is consistent with the study by Liu et al. [16].

In our study period, urban sprawl in China consistently showed spatial autocorrelation, which was also observed in Switzerland and the United States [48,49]. However, classical mathematical statistics and ESDA methods struggle to represent either the characteristics of spatial correlations or those of the time dimension [41]. In contrast, exploratory spatiotemporal data analysis (ESTDA) can more effectively integrate the temporal dimension with the spatial dimension for spatiotemporal interaction visualization, which has been used in disease prevention and control, air pollution, resource allocation, land-use, land-cover changes, and other fields [50–52].

In addition, among studies of drivers of urban sprawl, our results are largely consistent with existing research. Zhang et al. [29] argued that economic dimensions have a greater impact on urban sprawl in China compared to other dimensions. Li and Li [8] evaluated the socioeconomic drivers of urban sprawl in China, arguing that urban sprawl is closely related to urban population density, per capita GDP, and industrial structure. Their findings are similar to those in this study. The Geodetector method has advantages over principal component analysis and geographically weighted regression methods in determining the influence of explanatory variables [53]. In addition to urban-related studies, Geodetector models have received much attention in several other areas, such as vegetation change, land pollution, and medical issues, due to their superior performance [54–56].

*5.2. Policy Implications*

This study has some important and direct theoretical and practical policy implications. First, rational control of urban growth requires the development of scientifically sound urban development plans. Ineffective urban planning has often been deemed one of the important factors contributing to urban sprawl in Chinese cities [46,57]. Planning should be based on the development of the city itself, taking into account the differences among different regions and cities. Suggested actions include actively carrying out urban regeneration activities, enhancing the development potential of urban internal space, making rational use of stock land, strengthening urban spatial constraints, and promoting urban sustainable development. Secondly, multiple factors driving urban sprawl should be considered and combined to formulate comprehensive policies that can control urban sprawl. For instance, population growth causes an increase in housing demand, which promotes the migration of the real estate industry to the periphery of cities. In addition, convenient urban transportation conditions promote the outward diffusion of the population, which accelerates urban sprawl. Therefore, in order to control urban sprawl, multiple factors should be considered comprehensively; economic development should be restrained in areas with excessive expansion of urban space, and urban boundaries should be appropriately controlled to effectively restrain urban sprawl.

*5.3. Research Limitations and Prospects*

Overall, this study generally achieved the research objectives, measured urban sprawl in China, obtained the spatial and temporal patterns of urban sprawl in China, and defined its main influencing factors. However, the study still has some shortcomings. Due to the ambiguity of the concept, data inconsistency, and the subjectivity of index selection, this paper has some limitations that need to be further studied. Firstly, the fuzzy concept of urban sprawl and that of its process make it difficult to measure. A single-index measure cannot explain the multidimensional nature of urban sprawl [57]. Many sub-indicators are highly correlated, which increases the risk of capturing the causes and consequences of urban sprawl rather than describing the phenomenon of urban sprawl itself [8]. Secondly, the accuracy and inconsistency of the data presented challenges in this study. The research period (2000–2017) was limited to the years in which DMSP-OLS and NPP-VIIRS data were available. Although the two sets of data were integrated with reference to previous studies [36,37], and attempts were made to reduce the data differences caused by different spatial and radiative resolutions, it is still impossible to completely eliminate differences caused by different data sources. Due to the different geographical environments of each city, the great difference in urban morphology, the large differences in city size, and the low spatial resolution of night light images, the built-up area range extracted by the change detection method inevitably has some deviations. Urban sprawl is a complex process of urban development, and more detailed research on urban sprawl, with a longer time series, should be performed. In addition, we suggest expanding the research scale to understand urban sprawl in the United States, Europe, and the rest of the world.

## 6. Conclusions

(1) China experienced moderate-to-high urban sprawl development from 2000 to 2017. In terms of spatial distribution, the USI was high in the northwest and low in the southeast. Changes in the USI increasingly appeared in prefecture-level cities located at boundaries between provinces.

(2) The local spatial stability of the USI gradually decreased from southeast to northwest and northeast. There was no evident spatiotemporal transformation, strong spatial cohesion, relatively stable spatial pattern, or specific transfer inertia of the USI.

(3) The gross regional product (GRP) of the tertiary industry, the general GRP, and investment in real estate development were the most important factors affecting the sprawl in cities at the prefecture level and above in China.

**Author Contributions:** Conceptualization, X.Z. and J.P.; methodology, X.Z. and J.P.; software, X.Z.; validation, X.Z.; formal analysis, X.Z.; investigation, X.Z.; resources, X.Z.; data curation, X.Z.; writing—original draft preparation, X.Z.; writing—review and editing, X.Z. and J.P.; visualization, X.Z.; supervision, J.P.; project administration, J.P.; funding acquisition, J.P. All authors have read and agreed to the published version of the manuscript.

**Funding:** This study was funded by the National Natural Science Foundation of China (no. 42071216).

**Institutional Review Board Statement:** Not applicable.

**Informed Consent Statement:** Not applicable.

**Data Availability Statement:** Nighttime Light Data were collected from the National Oceanic and Atmospheric Administration/National Geophysical Data Center: https://www.ngdc.noaa.gov/eog/download.html (accessed on 20 March 2020). LandScan Global Population Data were collected from Oak Ridge National Laboratory's website: https://landscan.ornl.gov/landscan-datasets (accessed on 27 March 2020). Administrative Boundary Data were collected from the National Geomatics Center of China (http://www.ngcc.cn/ngcc/ (accessed on 20 March 2020)).

**Conflicts of Interest:** The authors declare no conflict of interest.

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
