# Peer review of "Spatiotemporal Pattern and Driving Factors of Urban Sprawl in China"

_land, doi:10.3390/land10111275_

Round 1

Reviewer 1 Report

This study aims at the quantification of urban sprawl in 343 Chinese cities based on night light data and LandScan population data at administrative spatial units. As a novel aspect and compared to other studies, relevant intra-urban differences are considered based on the proposed approach. In addition, this work aims at the identification of driving factors of urban sprawl in China.

In general, the proposed procedure appears feasible and the employed methods seem to be suitable for solving the problem posed. The style of writing is mostly appropriate. However, there are some crucial issues with this paper from my point of view.

While the first part of the paper on measuring urban sprawl and its spatio-temporal pattern reads quite promising, the analysis and presentation of influencing factors is insufficient from my point of view. While it is easily possible to bring together methods and results of sections 3 and 4 and to understand implications, section 5 leaves the reader lost and confused. Section 5 mixes up parts belonging to section 2 (data source), section 3 (methods), results, and discussion. From my point of view, this part must be reworked fundamentally.

In addition, a real discussion of the results is missing throughout. Sections 3 & 4 of the manuscript are presented in a rather technical & methodological manner, a discussion of the methodology along with a relation of the proposed approach to other studies is sorely missing. This applies in particular to the discussion of the results with regard to influencing factors, where there is no reference (What driving factors were identified by other studies? What conclusions do other authors draw?).

For these reasons, I think the paper is not ready for publication in its present form and urgently needs to be revised properly. In general, my advice for this work is to either focus on measuring urban sprawl and its spatio-temporal patterns or otherwise suitable embed and improve the analysis on influential factors. In both cases you need to rework and enrich the discussion fundamentally from my point of view.

Minor comments:

  • Line 33-37: These two sentences basically state the same
  • Line 41-43: I don’t understand the relation of increasing cost of public infrastructure construction and the ghost city phenomenon? The paper you cite is on farmland conversion to built-up areas and has nothing to do with ghost cities, I recommend more specific references:

Jin, X, Long, Y, Sun, W, Lu, Y, Yang, X, & Tang, J (2017): Evaluating cities’ vitality and identifying ghost cities in China with emerging geographical data. Cities 63, 98–109.

Leichtle, T, Lakes, T, Zhu, X X, & Taubenböck, H (2019): Has Dongying developed to a ghost city? Evidence from multi-temporal population estimation based on VHR remote sensing and census counts. Computers, Environment and Urban Systems 78, 101372.

  • Line 47-48: Please provide a reference here
  • Line 94-97: Please provide a reference here
  • Line 164-186:
    1. What is “complex” about the urban sprawl index? I also wonder about the “complexity” of land sprawl and population sprawl? I believe the terminology “complexity” does not well reflect your approach considering intra-urban differences.
    2. I am also a bit confused with the terminology of your land sprawl index (USCA). To my understanding, speaking of low density in the context of land cover implies low built-up density. However, your index refers to population density, which is different. Please use a clearer terminology here.
  • Line 187, line 191: Please introduce abbreviations ESTDA and LISA.
  • Line 226-227: How do you define the intermediate level? Can you provide other examples for comparison, would be interesting to compare for example the US, Europe, etc.

Author Response

Dear Reviewers,

Thank you for your letter and your comments on our manuscript entitled Spatio-temporal pattern and driving factors of urban sprawl in China (Manuscript ID: land-1454026). Those comments are all valuable and have been very helpful for revising and improving our paper and providing important guidance for our research. We have studied the comments carefully and have made corrections accordingly, which we hope will be approved. The revised portions are marked in revised mode in the paper. The main corrections to the paper and the responses to the reviewers’ comments are as follows:

While the first part of the paper on measuring urban sprawl and its spatio-temporal pattern reads quite promising, the analysis and presentation of influencing factors is insufficient from my point of view. While it is easily possible to bring together methods and results of sections 3 and 4 and to understand implications, section 5 leaves the reader lost and confused. Section 5 mixes up parts belonging to section 2 (data source), section 3 (methods), results, and discussion. From my point of view, this part must be reworked fundamentally.

Response: Thank you for raising this important question. The main content of this paper is to propose a method to measure the spatialization of urban sprawl using remote sensing data, and apply it to the measurement of urban sprawl and the analysis of temporal and spatial pattern in China. Limited by the length of the paper and the huge differences among cities, the analysis of the influencing factors of urban sprawl is relatively weak. In the revised paper, we added the analysis and introduction of driving factors, added the introduction of driving factors in the introduction, and added the discussion of driving factors in the discussion part. In addition, as the reviewer said, section 5 leaves the reader lost and confused. So, the section 5 was reorganized. We divide section 5 into several parts and classify them into data source, methods, results, and discussion sections. 

The revised introduction is as follows:

As research goes deeper, the study of the intrinsic mechanism of urban sprawl also becomes a research hotspot. In the long run, identifying the drivers of urban sprawl is important to promote urban development policies. According to the existing literature, the driving force of urban sprawl can be explored from the following aspects: politics, economy, sociology, environment [3,8,25]. Urban sprawl in western countries is mainly caused by market forces, consumer preferences, public subsidies, and land-use regulation are considered the main drivers of urban sprawl in western countries [26]. Burchfield et al. [27] researched that ground water availability, temperate climate, rugged terrain, decentralized employment, early public transport infrastructure, uncertainty about metropolitan growth, and unincorporated land in the urban fringe all increase sprawl. Research by Pirotte and Madre [28] shows that the French urban development model is highly dependent on the density of surrounding farms and their ability to provide facilities. Because of its very different institutional context, the causes of urban sprawl in China contain many aspects, among which government regulation factors, economic factors, social factors predominate [3,29]. And, most of the related research is qualitative research, cases of quantitative and spatialized studies on large scale scales are extremely rare. Yue et al. [24] discussed the impacts of farmland preservation policy, policy on population, and urban planning on urban sprawl in Hangzhou City. Overall, the related research on urban sprawl mostly focuses on a single city (or urban agglomeration) on a small scale [30-32], ignoring the analysis of spatial-temporal differentiation and spatial-temporal correlation characteristics of urban sprawl from the perspective of a large scale.

The discussion of the drivers of urban sprawl is revised as follows:

Meanwhile, among studies of drivers of urban sprawl, our results are largely consistent with existing research. Zhang et al. [29] argued that economic dimensions have a greater impact on urban sprawl in China. Li and Li [8] evaluates the socio-economic drivers of urban sprawl in China, arguing that urban sprawl is closely related to urban population density, per capita GDP and industrial structure. This is all similar to our findings. And Geodetector method had more advantages over the principal component analysis and geographically weighted regression methods in determining the influence of explanatory variables [53]. In addition to urban related studies, Geodetector models have received much attention for various areas, such as vegetation change, land pollutions, and medical issues, for their superiority [54-56].

The table of contents after revising the article structure is as follows:

  1. Introduction
  2. Data Source

2.1. Data

2.2. Data processing

  1. Methods

3.1. Urban sprawl index

3.2. Exploratory Spatiotemporal Data Analysis (ESTDA)

3.3. Indicator system of driving factors

3.4. Geodetector

  1. Results

4.1. General characteristics of urban sprawl

4.2. Spatial and temporal patterns of USI 

4.3. Driving factors of urban sprawl

4.3.1. Factor detection analysis

4.3.2. Interactive detection analysis

  1. Discussion

5.1. Comparison with other studies

5.2. Policy enlightenment

5.3. Research deficiencies and prospects

  1. Conclusions

In addition, a real discussion of the results is missing throughout. Sections 3 & 4 of the manuscript are presented in a rather technical & methodological manner, a discussion of the methodology along with a relation of the proposed approach to other studies is sorely missing. This applies in particular to the discussion of the results with regard to influencing factors, where there is no reference (What driving factors were identified by other studies? What conclusions do other authors draw?).

Response: Thank you for raising the question. We have added a new section 5.1 Comparison with other studies, which includes discussion of urban sprawl index, ESTDA and influencing factor analysis, and the research methods and research results are compared with other studies. The added content is as follows:

5.1. Comparison with other studies

Because of the complex and multi-dimensional nature of the urban sprawl problem, most scholars measure them by constructing indicators. Despite the simplicity of single index measure calculation, there is limited ability to characterize urban sprawl relatively completely [47]. The combination of multiple indicators has higher relevance and in turn increases the risk of inclusion of causes and consequences of sprawl. In this paper, we use the indicators of the population and area aspects of the phenomenon of urban sprawl, and try to consider the multi-dimensional nature of urban sprawl. And the urban sprawl index referred to in this paper can subdivide the internal differences of the city, no longer regard the city as a whole, making the measurement of urban sprawl more reasonable. The results show that geographical differences in urban sprawl are significant, the extent of urban sprawl in the central and western regions was greater than that in the eastern and northeastern regions, which was consistent with the study of Liu et al. [16].

In our study, urban sprawl in China always has spatial autocorrelation, which was also been confirmed in Switzerland and the United States [48,49].But classical mathematical statistics and ESDA methods either struggle to represent the characteristics of spatial correlations or that of the time dimension [41]. In contrast, exploratory spatiotemporal data analysis (ESTDA) can more effectively integrate the temporal dimension with the spatial dimension for spatiotemporal interactive visualization, which has been used in disease prevention and control, air Pollution, resource allocation, land-use and land-cover changes, and otherfields [50-52].

Meanwhile, among studies of drivers of urban sprawl, our results are largely consistent with existing research. Zhang et al. [29] argued that economic dimensions have a greater impact on urban sprawl in China. Li and Li [8] evaluates the socio-economic drivers of urban sprawl in China, arguing that urban sprawl is closely related to urban population density, per capita GDP and industrial structure. This is all similar to our findings. And Geodetector method had more advantages over the principal component analysis and geographically weighted regression methods in determining the influence of explanatory variables [53]. In addition to urban related studies, Geodetector models have received much attention for various areas, such as vegetation change, land pollutions, and medical issues, for their superiority [54-56].

For these reasons, I think the paper is not ready for publication in its present form and urgently needs to be revised properly. In general, my advice for this work is to either focus on measuring urban sprawl and its spatio-temporal patterns or otherwise suitable embed and improve the analysis on influential factors. In both cases you need to rework and enrich the discussion fundamentally from my point of view.

Response: We are grateful for the suggestion, and serious consideration of the observations. The influencing factors section in the original paper is indeed weak. We believe that the focus of this paper is to measure urban sprawl in China and explore its spatio-temporal pattern, and influencing factors are not. But for the completeness of the paper structure, we have maintained this section in the original article. We have simplified the analysis regarding influencing factors in our revised manuscript. And we are also considering removing this section to highlight the focus of this study, making the article more refined. We hope that in the future research, we can make an in-depth and detailed analysis on the driving mechanism of urban sprawl.

Minor comments:

Line 33-37: These two sentences basically state the same

Response: Thank you for raising this question. We deleted repeated sentences and checked the full text carefully.

Line 41-43: I don’t understand the relation of increasing cost of public infrastructure construction and the ghost city phenomenon? The paper you cite is on farmland conversion to built-up areas and has nothing to do with ghost cities, I recommend more specific references: 

Jin, X, Long, Y, Sun, W, Lu, Y, Yang, X, & Tang, J (2017): Evaluating cities’ vitality and identifying ghost cities in China with emerging geographical data. Cities 63, 98–109.

Leichtle, T, Lakes, T, Zhu, X X, & Taubenböck, H (2019): Has Dongying developed to a ghost city? Evidence from multi-temporal population estimation based on VHR remote sensing and census counts. Computers, Environment and Urban Systems 78, 101372.

Response: Thank you for raising this important question. Our expression in the original text is not complete and accurate. We believe that the increase in the cost of public infrastructure may lead to temporary infrastructure imperfections that cannot meet the needs of population and industrial agglomeration. On this basis it is highly likely to lead to the phenomenon of " empty city " and " ghost city ". For example, Kangbashi District in China was called a ghost town by western media due to the lack of various facilities and small resident population at the initial stage of construction. We corrected this sentence in our revised manuscript and added suitable references. The revised content is as follows:

The increase in the city scale will also increase the cost of public infrastructure construction, and the imperfection of infrastructure may lead to the phenomenon of "empty city" and "ghost city" [9,10], reducing the land-use efficiency.

We added the following references:

  1.   Jin X.;Long Y.;Sun W.; Lu Y.; Yang X.; Tang J. Evaluating cities' vitality and identifying ghost cities in china with emerging geographical data. Cities 2017. 63, 98–109. https://doi.org/10.1016/j.cities.2017.01.002.
  2.  Leichtle T.;Lakes T.;Zhu X. X.; H Taubenbck. Has dongying developed to a ghost city? - evidence from multi-temporal population estimation based on vhr remote sensing and census counts. Computers Environment and Urban Systems 2019, 78, 101372. https://doi.org/10.1016/j.compenvurbsys.2019.101372.

Line 47-48: Please provide a reference here

Response: Thanks for your suggestion, we have added the following references:

The definition of urban sprawl is the primary task of urban sprawl research [15].

  1.  Galster G.; Hanson R.; Ratcliffe M. R. Wrestling Sprawl to the Ground: Defining and Measuring an Elusive Concept. Housing Policy Debate2001, 12(4):681-717. https://doi.org/10.1080/10511482.2001.9521426.

Line 94-97: Please provide a reference here

Response: Thanks for your suggestion, we have added the following references:

Because of its very different institutional context, the causes of urban sprawl in China contain many aspects, among which government regulation factors, economic factors, social factors predominate [3,29].

  1.   Feng Y.; Wang X.; Du W.; Liu J.; Li Y. Spatiotemporal characteristics and driving forces of urban sprawl in China during 2003-2017.J. Cleaner Prod.2019, 241, 118061. https://doi.org/10.1016/j.jclepro.2019.118061.
  2.  Zhang X; Lu L; Ren Y.; Xu Y.; Zhang H. Spatiotemporal Evolution Pattern of Urban Sprawl in China and Its Influencing Factors,Economic Geography2021, 41(03):77-85. (in Chinese). https://doi.org/10.15957/j.cnki.jjdl.2021.03.008.

Line 164-186: What is “complex” about the urban sprawl index? I also wonder about the “complexity” of land sprawl and population sprawl? I believe the terminology “complexity” does not well reflect your approach considering intra-urban differences.

Response: Thank you for raising this important question. We have adopted the term "urban sprawl complexity", on the one hand because of the complexity of urban sprawl, and on the other hand, it is different from the conventional urban sprawl index (Such as the ratio of land growth rate to population growth rate). The urban sprawl index in this paper does not consider the population and area of a city as a whole, but to distinguish the difference in population distribution and land area occupation caused by the density difference within a city, and comprehensively considers both population and area. But it is likely to cause confusion, we delete the term "complexity" in the revised paper. See the revision for details.

I am also a bit confused with the terminology of your land sprawl index (USCA). To my understanding, speaking of low density in the context of land cover implies low built-up density. However, your index refers to population density, which is different. Please use a clearer terminology here.

Response: Thank you for raising this important question. This paper holds that if the proportion of population in low-density areas increases, or the proportion of population living area in low-density areas increases (this means that the density of urban low-density areas continues to decrease), it reflects the increase of urban sprawl. Therefore, land sprawl refers to that the area of low population density area exceeds that of high population density area. But this is really different from the meaning of low density in the context of land cover, so we modify the land sprawl index to the area sprawl index.

Line 187, line 191: Please introduce abbreviations ESTDA and LISA.

Response: We are grateful for the suggestion. The full name of EATDA is Exploratory Spatio-temporal Data Analysis. The full name of LISA is Local Indicators of Spatial Association. We added their full names in the revised manuscript.

Line 226-227: How do you define the intermediate level? Can you provide other examples for comparison, would be interesting to compare for example the US, Europe, etc.

Response: Thank you for raising this important question. We divided the urban sprawl index into 5 levels in this paper, and the mean of the urban sprawl index for all Chinese cities is located in the third level, so we think that overall, urban spread in China is at a moderate level. We have revised the statement of this sentence to make it clearer. Since the study area of this paper is in China, the measures of America, Europe, etc. were not taken, and it is also difficult to compare with others' studies in this regard. Thank you again for your suggestions. In the future, we will consider measures of urban sprawl in the USA, Europe and even globally. We added this outlook in the discussion section.

Furthermore, we have polished and consulted native English speakers for paper revision before the submission this time. We obtained assistance from MDPI English Editing Services for the revision of English grammar, spelling, sentence structure, and organization quality. This document certifies that the manuscript listed below was edited for proper English language, grammar, punctuation, spelling, and overall style by the highly qualified native English speaking editors at MDPI English Editing Services. The following is the English language editing certification.

Reviewer 2 Report

This article debates aspects related to urban sprawl complexity in China.

The abstract needs to be reorganized/ rephrased. It is not clear which is the main objective/ aim of this article. Also, it is not clearly mentioned the main conclusion.

Please improve the Introduction section by adding a couple of sentences mentioning which are the steps followed in this work to achieve the article objectives.

The method followed in this article is not clearly explained. You can improve this aspect by adding an explanatory graph.

The results should be better correlated with the article objectives. 

Discussions section needs also to be developed and better correlated with the objectives and results of this article. 

Overall, the article needs to be improved in terms of presentation quality and scientific soundness.

Major revision is required.

Author Response

Dear Reviewers,

Thank you for your letter and your comments on our manuscript entitled Spatio-temporal pattern and driving factors of urban sprawl in China (Manuscript ID: land-1454026). Those comments are all valuable and have been very helpful for revising and improving our paper and providing important guidance for our research. We have studied the comments carefully and have made corrections accordingly, which we hope will be approved. The revised portions are marked in revised mode in the paper. The main corrections to the paper and the responses to the reviewers’ comments are as follows:

This article debates aspects related to urban sprawl complexity in China. The abstract needs to be reorganized/ rephrased. It is not clear which is the main objective/ aim of this article. Also, it is not clearly mentioned the main conclusion.

Response: We are grateful for the suggestion. We reorganized and revised the abstract, emphasizing the main research objectives and conclusions of this study. The revised abstract is as follows:

Urban sprawl has become a key issue of global concern as a complex phenomenon related to abnormal urbanization. This study aims to measure urban sprawl in China and explore its Spatio-temporal patterns and driving factors Based on the 343 cities at the prefecture level and above in China from 2000 to 2017, this paper has adopted the remote sensing derived data to constitute the urban sprawl index (USI). The evolutionary trend and spatio-temporal pattern of urban sprawl in China were then analyzed using trend analysis and exploratory spatio-temporal data analysis, while Geodetector was applied to probe the factors driving the changes. The results show the following. â‘ There has been moderate or high urban sprawl development in China from 2000 to 2017. In terms of spatial distribution, the USI was high in the northwest China and low in the southeast China. â‘¡The local spatial stability of the USI gradually decreased from southeast to northwest and northeast. There is strong spatial dependence on the USI. No significant spatiotemporal transitions were observed in urban sprawl, and the spatial pattern was stable with strong spatial cohesion â‘¢The gross regional product (GRP) of tertiary industry, the general GRP, and investment in real estate development are the most important factors affecting the sprawl in cities at the prefecture level and above in China.

Please improve the Introduction section by adding a couple of sentences mentioning which are the steps followed in this work to achieve the article objectives.

Response: We are grateful for the suggestion. We reorganize the last paragraph of the introduction and clarify the main steps to achieve the objectives of this study. The revised content is as follows:

Based on the above analysis, refinedly identify the urban sprawl index of 343 prefecture-level and above cities in China as its research targets. This study includes three main parts of this study. First, we established the urban sprawl index capable of attending to the refinement differences that refine inner city areas by using nighttime light remote sensing data and LandScan population spatial distribution data, which was further used to measure urban sprawl in China between 2000 and 2017. Second, we analyzed the spatial-temporal patterns by introducing the Exploratory Spatial-Temporal Data Analysis Method (ESTDA). Third, we explored the drivers of urban sprawl. The ultimate goal of the third part was to provide decision-making references for the New Urbanization Construction and Regional Planning.

The method followed in this article is not clearly explained. You can improve this aspect by adding an explanatory graph.

Response: Thank you for raising the question. Due to space constraints, we have made a relatively concise description of the method in the original text. We added a flow process chart to improve this aspect in order to present the workflow and study methodology more clearly and intuitively. And in the introduction, method and discussion, the content of method interpretation is added. The added Figure is as follows:

Figure 1. flow process chart.

The results should be better correlated with the article objectives.

Response: We are grateful for the suggestion. We modified the results section to better relate to the objectives of this paper. For details, see the revised manuscript.

Discussions section needs also to be developed and better correlated with the objectives and results of this article.

Response: We are grateful for the suggestion. We modified the discussion section. In the revised manuscript, Section 5.1 discusses the findings, Section 5.2 suggests policy implications based on the results of the impact factor analysis, and section 5.3 discusses whether the research objectives were accomplished and the inadequacies that exist. The revised discussion is as follows:

  1. Discussion

Unlike the process of urbanisation in western developed countries, which is largely market driven, urbanisation in China is more often dominated by the government [45,46]. Local governments intervene in the spatial allocation of elements, which in turn determines the formation and development of urban spatial shape, through the regulation of land system and household registration system. Because of the " finance of land " and the pursuit of an expanded scale of urban space [46], making land urbanisation significantly faster than population urbanisation has become the norm in most cities in China.

5.1. Comparison with other studies

Because of the complex and multi-dimensional nature of the urban sprawl problem, most scholars measure them by constructing indicators. Despite the simplicity of single index measure calculation, there is limited ability to characterize urban sprawl relatively completely [47]. The combination of multiple indicators has higher relevance and in turn increases the risk of inclusion of causes and consequences of sprawl. In this paper, we use the indicators of the population and area aspects of the phenomenon of urban sprawl, and try to consider the multi-dimensional nature of urban sprawl. And the urban sprawl index referred to in this paper can subdivide the internal differences of the city, no longer regard the city as a whole, making the measurement of urban sprawl more reasonable. The results show that geographical differences in urban sprawl are significant, the extent of urban sprawl in the central and western regions was greater than that in the eastern and northeastern regions, which was consistent with the study of Liu et al. [16].

In our study, urban sprawl in China always has spatial autocorrelation, which was also been confirmed in Switzerland and the United States [48,49].But classical mathematical statistics and ESDA methods either struggle to represent the characteristics of spatial correlations or that of the time dimension [41]. In contrast, exploratory spatiotemporal data analysis (ESTDA) can more effectively integrate the temporal dimension with the spatial dimension for spatiotemporal interactive visualization, which has been used in disease prevention and control, air Pollution, resource allocation, land-use and land-cover changes, and otherfields [50-52].

Meanwhile, among studies of drivers of urban sprawl, our results are largely consistent with existing research. Zhang et al. [29] argued that economic dimensions have a greater impact on urban sprawl in China. Li and Li [8] evaluates the socio-economic drivers of urban sprawl in China, arguing that urban sprawl is closely related to urban population density, per capita GDP and industrial structure. This is all similar to our findings. And Geodetector method had more advantages over the principal component analysis and geographically weighted regression methods in determining the influence of explanatory variables [53]. In addition to urban related studies, Geodetector models have received much attention for various areas, such as vegetation change, land pollutions, and medical issues, for their superiority [54-56].

5.2. Policy enlightenment

Correspondingly, there are some important and straightforward policy implications for theory and practice, as follows: First, rational control of urban growth requires the development of scientifically sound urban development plans. Powerless urban planning often has been deemed as one of the important factors contributing to urban sprawl in Chinese cities [46,57]. Planning should be based on the development of the city itself, taking into account the differences of different regions and cities. Actively carry out urban regeneration activities, develop the development potential of urban internal space, make rational use of stock land, strengthen urban spatial constraints, and promote urban sustainable development. Secondly, multiple factors driving urban sprawl should be considered and combined to formulate comprehensive policies that can control urban sprawl. For instance, population growth causes the increase of housing demand, which promotes the development of the real estate industry to the periphery of cities. In addition, convenient urban transportation conditions promote the outward diffusion of population, which speeds up urban sprawl. Therefore, in order to control urban sprawl, multiple factors should be considered comprehensively, and economic development should be refused by excessive expansion of urban space, and urban boundary should be controlled reasonably to effectively restrain urban sprawl.

5.3. Research deficiencies and prospects

Overall, this paper basically achieves the research objectives, completes the measures of urban sprawl in China, obtains the spatial and temporal pattern of urban sprawl in China, and defines its main influencing factors. But there are still some shortcomings in the study. Due to the fuzziness of concept, inconsistency of data and the subjectivity of index selection, this paper also has some limitations, which need to be further studied. Firstly, the fuzzy concept of urban sprawl and the of its process make it difficult to measure urban sprawl. A single index measure cannot explain the multidimensional nature of  urban sprawl [57]. The combination of many sub-indicators is highly correlated, which increases the risk of containing the causes and consequences of urban sprawl rather than describing the phenomenon of urban sprawl itself [8]. Secondly, the accuracy and inconsistency of the data have brought a test to this study. The research period of this paper from 2000 to 2017 just spans the availability period of DMSP-OLS and NPP-VIIRS data. Although this paper integrates the two kinds of data with reference to previous studies [36,37], and tries to reduce the data differences caused by different spatial and radiative resolutions, it is still impossible to completely eliminate the differences caused by different data sources. Due to the different geographical environments of each city, the great difference in urban morphology, the huge difference in city size, and the low spatial resolution of night light images, the built-up area range extracted by the mutation detection method inevitably have deviation. Urban sprawl is a complex process of urban development, more detailed and longer time series research on urban sprawl could be paid more attention to. In addition, we consider expanding the research scale to understand the urban sprawl in the United States, Europe and even the world.

Overall, the article needs to be improved in terms of presentation quality and scientific soundness. Major revision is required.

Response: We are grateful for the suggestion. We have carefully revised all the above questions. We adjusted the organizational structure of the article, made appropriate modifications to all parts of the article, and added a work flow chart to make the article structure more reasonable and scientific. In order to improve the presentation quality, we turned to MPDI language editing service to revise English grammar, spelling, sentence structure and organization quality. The English Editing Certificate is as follows:

Reviewer 3 Report

Aim of the work is to adopt the remote sensing derived data to constitute the urban sprawl complexity index for the spatio-temporal pattern and driving factors of urban sprawl complexity in China. The work is structured into a well content's sequences, but some suggestions are provided in order to improve the work:

  • Introduction: contextualize better some statemets reguarding the lack of specific-related research on urban sprawl. In order words, could be useful if a separation were made between international studies on uncontrolled urban sprawl and those conducted in China. To say that in the literature of the sector there is a lack of research that has to do with the extra-urban scale, ignoring the analysis of spatial-temporal differentiation and spatial-temporal correlation characteristics of urban sprawl from the perspective of a large scale, and that there is no in -depth analysis and discussion on the influencing factors behind the urban sprawl phenomenon, especially the spatial factors, appears incorrect if the reference scope is not expressed. These statements may in fact be true for Chinese literature but not for American or Italian literature (for example), so it is good to clarify these passages. Furthermore, are necessary some brief lines that highlight how the sustainable development solutions such as the urban regeneration with all his complex issues have been raised within the urban sprawl research. On this, some references could be the following ones:
    • Morano, P., Guarini, M. R., Sica, F., & Anelli, D. (2021, September). Ecosystem Services and Land Take. A Composite Indicator for the Assessment of Sustainable Urban Projects. In International Conference on Computational Science and Its Applications (pp. 210-225). Springer, Cham.
    • Deal, B., & Schunk, D. (2004). Spatial dynamic modeling and urban land use transformation: a simulation approach to assessing the costs of urban sprawl. Ecological Economics51(1-2), 79-95.
    • Zambon, I., Benedetti, A., Ferrara, C., & Salvati, L. (2018). Soil matters? A multivariate analysis of socioeconomic constraints to urban expansion in Mediterranean Europe. Ecological Economics146, 173-183.
  • Data source: add the type of data related to the source
  • Methods: your proposed index is a novelty? which are the main references from which it derives if not?Explain.

Author Response

Dear Reviewers,

Thank you for your letter and your comments on our manuscript entitled Spatio-temporal pattern and driving factors of urban sprawl in China (Manuscript ID: land-1454026). Those comments are all valuable and have been very helpful for revising and improving our paper and providing important guidance for our research. We have studied the comments carefully and have made corrections accordingly, which we hope will be approved. The revised portions are marked in revised mode in the paper. The main corrections to the paper and the responses to the reviewers’ comments are as follows:

Aim of the work is to adopt the remote sensing derived data to constitute the urban sprawl complexity index for the spatio-temporal pattern and driving factors of urban sprawl complexity in China. The work is structured into a well content's sequences, but some suggestions are provided in order to improve the work:

Introduction: contextualize better some statemets reguarding the lack of specific-related research on urban sprawl. In order words, could be useful if a separation were made between international studies on uncontrolled urban sprawl and those conducted in China. To say that in the literature of the sector there is a lack of research that has to do with the extra-urban scale, ignoring the analysis of spatial-temporal differentiation and spatial-temporal correlation characteristics of urban sprawl from the perspective of a large scale, and that there is no in-depth analysis and discussion on the influencing factors behind the urban sprawl phenomenon, especially the spatial factors, appears incorrect if the reference scope is not expressed. These statements may in fact be true for Chinese literature but not for American or Italian literature (for example), so it is good to clarify these passages.

Response: We are grateful for the suggestion. In the original introduction section, the research review did not clearly distinguish the studies in China from those in other countries, nor did it indicate the geographical range to which they apply. Therefore, we made a clear distinction in our revised manuscript between the differences in research progress in China and Western countries. The geographical range to which it applies is also clearly stated when citing relevant literature. In addition, we have conducted more in-depth analysis and discussion on the literature review. The revised introduction is as follows:

  1. Introduction

Urban sprawl has been a topic of controversy [1]. Urban sprawl is a special and out-of-control urbanization development trend currently facing all countries in the world. Urban sprawl refers to an unsustainable spatial expansion pattern that the city shows in the process of development: random and unplanned, scattered and discontinuous, single land use mode, extremely dependent on transportation for travel, and land use contradiction tends to be severe [2]. With industrialization advancing, the improvement of highway transport facilities, and the popularity of motor vehicles, suburban areas rapidly develop, urban centers tend to decline, and the sprawl has gradually become a prevalent urban development situation in developed western countries [3]. Since the initiation of the “Reform and Opening-Up” policy, urbanization is advancing extremely rapidly in China [4]. In 1978, the urbanization rate calculated as the resident population was only 18%, and this number has increased to 60.6% by 2019, which requires the larger urban capacity to accommodate the dramatically increasing urban industry and population. In 2019, the per capita construction land in Chinese cities was as high as 200 m2, which was much higher than that in developed countries. The outward expansion of the city has become the "normal" of many cities, and the blind expansion of the built-up areas can be seen everywhere [5]. Urban sprawl usually brings adverse effects to the development of the city, such as traffic congestion, environmental pollution, social space differentiation, etc.[6-8]. In addition, the outward expansion of the city often comes at the expense of the quantity and quality of cultivated land. The increase in the city scale will also increase the cost of public infrastructure construction, and the imperfection of infrastructure may lead to the phenomenon of "empty city" and "ghost city" [9,10], reducing the land-use efficiency. Furthermore, this is also significant for sustainable development research. Such as The insufficient ecosystem services related to phenomena of uncontrolled urban expansion in the cities force many countries far away from achieving sustainable development goals [11]. The economic impacts resulting from urban land use transformation has become complex and acrimonious [12], and Urban expansion and socioeconomic transformations may negatively impact environmental quality and functions of peri-urban landscapes [13]. The continuous emergence of the above drawbacks has attracted the general attention of the government, the public, and scholars [14].

Many scholars have carried out in-depth research on the issues related to urban sprawl, mainly from the concept definition, evaluation methods, internal mechanism, regulatory measures, and the ecological and environmental effects of urban sprawl. The definition of urban sprawl is the primary task of urban sprawl research [15]. Although scholars have an in-depth understanding of urban sprawl, due to the complexity and multi-dimensions of urban sprawl, there has not yet been a unified definition of urban sprawl. At present, the existing related concepts are mainly defined from the manifestations, formation characteristics, and impacts of urban sprawl [16]. From the existing literature, the sprawl in western countries is essentially the expansion of cities beyond urban boundaries to extra-urban, accompanied by low density and transport dependency [15,17]. However, because developing countries are in a stage of rapid urbanization, they are distinguished from western countries in terms of population density, urban-rural structure, land system, and socioeconomic development level. China is no exception. Based on the definition of sprawl in western countries, the definition in China is more specific and focuses on describing the manifestations of one or a certain type of urban sprawl. Yue et al. [18] defined urban sprawl as a low-density type of urban expansion occurred beyond the urban built-up area, including low density edge-growth or leapfrog growth such as industrial development zones or college towns. But many scholars confuse urban sprawl with urban expansion, which is distinguished in this paper. Urban expansion characterizes the pursuit of "absolute scale" by depicting the spatial-temporal changes of urban land use. In contrast, urban sprawl describes the changing development of cities in terms of three dimensions: low density, scale-up, and negative impacts, which can reflect imbalances and inadequate development. Therefore, it is biased to think that urban expansion is urban sprawl.

The vague concept measures urban sprawl as a difficult problem. Measures of urban sprawl have diverged as the concept evolves. In particular, China and the West are distinguished in the concept of urban sprawl, and hence in measures. Western scholars use to calculating the urban sprawl index by applying a quantitative metric in combination with space. They used a single indicator measure approach early, while with increasing knowledge of sprawl and the region under study, a multi indicator approach was gradually adopted to construct a comprehensive urban sprawl index. The study of urban sprawl in China started relatively late, and the measures were more fumbled. Urban sprawl in China is mostly based on single indicator method measures. Such as Lang and Lefurgy [19] measure urban sprawl with population density indicators in the United States. Taiwo [20] has constructed the Urban Sprawl Index, which is the ratio of the area growth rate of built-up area to the population growth rate (elastic coefficient) to measure the urban sprawl in Nigeria.. The above studies fail to distinguish the distribution of even or concentrated populations within a city. Scholars have proposed urban sprawl indices that take into account differences in population distribution within cities. Lopez and Hynes [21] and Fallah et al. [22] constructed the sprawl index based on the spatial distribution of population density within the city and subtracted the population proportion of low-density areas within the city from that of high-density areas to reflect the extent of urban sprawl in the United States. Li and Li [8] used the difference between the urban area growth rate and the population growth rate to measure urban sprawl in China. If the proportion of low population density area is larger, it means that the intensity of land use is low and the degree of urban sprawl is higher. After that, scholars have improved the above indicators. Try to make the measurement result of urban sprawl reflect the spatial details in the city more and be less disturbed by the abnormal distribution of the local population. Based on the analysis of urban land sprawl and urban population sprawl, Qin et al. [23] constructed a new urban sprawl index to study urban sprawl in China.

After identifying an urban sprawl measurement method, Chinese and Western scholars have gradually focused their research attention on the spatial characteristics presented by urban sprawl. In general, spatial characteristics are relatively easily observed, such as, low-density development, leapfrog or scattered development, and poor accessibility, etc. [16,18]. Later, scholars add a temporal dimension on this basis and explore the spatiotemporal changing characteristics of urban sprawl, which deepen our research to some extent. Nazarnia et al. [24] explored the spatial pattern of urban sprawl in Montreal, Quebec City, and Zurich from 1951 to 2011. Feng et al. [3] investigated the spatial correlations and spatial distribution patterns of urban sprawl in China by utilizing Global Moran's I (GMI) and Local Moran's I (LMI) The study of spatiotemporal changes is relatively shallow and only remains superficial, and the study of the spatiotemporal pattern of urban sprawl receives little attention.

As research goes deeper, the study of the intrinsic mechanism of urban sprawl also becomes a research hotspot. In the long run, identifying the drivers of urban sprawl is important to promote urban development policies. According to the existing literature, the driving force of urban sprawl can be explored from the following aspects: politics, economy, sociology, environment [3,8,25]. Urban sprawl in western countries is mainly caused by market forces, consumer preferences, public subsidies, and land-use regulation are considered the main drivers of urban sprawl in western countries [26]. Burchfield et al. [27] researched that ground water availability, temperate climate, rugged terrain, decentralized employment, early public transport infrastructure, uncertainty about metropolitan growth, and unincorporated land in the urban fringe all increase sprawl. Research by Pirotte and Madre [28] shows that the French urban development model is highly dependent on the density of surrounding farms and their ability to provide facilities. Because of its very different institutional context, the causes of urban sprawl in China contain many aspects, among which government regulation factors, economic factors, social factors predominate [3,29]. And, most of the related research is qualitative research, cases of quantitative and spatialized studies on large scale scales are extremely rare. Yue et al. [24] discussed the impacts of farmland preservation policy, policy on population, and urban planning on urban sprawl in Hangzhou City. Overall, the related research on urban sprawl mostly focuses on a single city (or urban agglomeration) on a small scale [30-32], ignoring the analysis of spatial-temporal differentiation and spatial-temporal correlation characteristics of urban sprawl from the perspective of a large scale.

Based on the above analysis, refinedly identify the urban sprawl index of 343 prefecture-level and above cities in China as its research targets. This study includes three main parts of this study. First, we established the urban sprawl index capable of attending to the refinement differences that refine inner city areas by using nighttime light remote sensing data and LandScan population spatial distribution data, which was further used to measure urban sprawl in China between 2000 and 2017. Second, we analyzed the spatial-temporal patterns by introducing the Exploratory Spatial-Temporal Data Analysis Method (ESTDA). Third, we explored the drivers of urban sprawl. The ultimate goal of the third part was to provide decision-making references for the New Urbanization Construction and Regional Planning.

Furthermore, are necessary some brief lines that highlight how the sustainable development solutions such as the urban regeneration with all his complex issues have been raised within the urban sprawl research. On this, some references could be the following ones:

Morano, P., Guarini, M. R., Sica, F., & Anelli, D. (2021, September). Ecosystem Services and Land Take. A Composite Indicator for the Assessment of Sustainable Urban Projects. In International Conference on Computational Science and Its Applications (pp. 210-225). Springer, Cham.

Deal, B., & Schunk, D. (2004). Spatial dynamic modeling and urban land use transformation: a simulation approach to assessing the costs of urban sprawl. Ecological Economics, 51(1-2), 79-95.

Zambon, I., Benedetti, A., Ferrara, C., & Salvati, L. (2018). Soil matters? A multivariate analysis of socioeconomic constraints to urban expansion in Mediterranean Europe. Ecological Economics, 146, 173-183.

Response: We are grateful for the suggestion. In the introduction, we emphasize the importance of urban sprawl research to sustainable development, and add relevant literature. In addition, in the discussion, we refer to the active implementation of urban renewal, the continuous development of corresponding policy measures to ameliorate developing problems, the exploitation of the potential for inner urban space development, the rational use of stored land, the strengthening of urban spatial constraints, and the promotion of urban sustainability. The revised Introduction is as follows:

Furthermore, this is also significant for sustainable development research. Such as The insufficient ecosystem services related to phenomena of uncontrolled urban expansion in the cities force many countries far away from achieving sustainable development goals [11]. The economic impacts resulting from urban land use transformation has become complex and acrimonious [12], and Urban expansion and socioeconomic transformations may negatively impact environmental quality and functions of peri-urban landscapes [13].

  1.  Morano P.; Guarini M. R.; Sica F.;Anelli D. Ecosystem Services and Land Take. A Composite Indicator for the Assessment of Sustainable Urban Projects, In International Conference on Computational Science and Its Applications, Cagliari, Italy, September 12, 2021; pp. 215-225. https://doi.org/10.1007 / 978-3-030-86979-3_16
  2.  Deal B.; Schunk D. Spatial dynamic modeling and urban land use transformation: a simulation approach to assessing the costs of urban sprawl. Ecological Economics 2004, 51(1-2), 79-95.https://doi.org/10.1016/j.ecolecon.2004.04.008.
  3.  Zambon I.; Benedetti A.; Ferrara C.; Salvati L. Soil matters? A multivariate analysis of socioeconomic constraints to urban expansion in Mediterranean Europe. Ecological Economics2018., 146, 173-183.https://doi.org/10.1016/j.ecolecon.2017.10.015

Data source: add the type of data related to the source

Response: We are grateful for the suggestion. Nighttime Light Database, LandScan Global Population Database are grid data. Administrative boundary data is vector data. All the driving forces data used in this paper are selected from China City Statistical Yearbook, which is statistical data. We added the type of data related to the source in the revised draft.

Methods: your proposed index is a novelty? which are the main references from which it derives if not? Explain.

Response: Thank you for raising the question. This paper refers to the improved urban sprawl index (USI) of Qin et al [23]. This index integrates the characteristics of low-density expansion of urban sprawl, integrating both urban populations and land area, to try to account for the multidimensional nature of urban sprawl. If the proportion of the population in low-density areas of a city increases, or the living area proportion of the population in low-density areas increases (this means that the density of urban low-density areas continues to decrease), both reflect the rising degree of complex urban sprawl. Compared with the commonly used average density method to measure urban sprawl, USI considers the detailed differences of urban internal regions. It does not regard the city as a homogeneous area, thus avoiding the influence of areas with abnormally high or low density in the city on the overall density of the city, so it is more reasonable to estimate the urban sprawl. Before that, Lopez et al. [21] and Fallah et al. [22] Constructed the spread index based on the spatial distribution of population density within the city. Subtracting the population share of low-density areas within cities from that of high-density areas reflects the spread of the city. The larger proportion of low population density areas indicates low land use intensity and increased urban sprawl.

  1.  Lopez R.; Hynes H. P. Sprawl in the 1990s: measurement, distribution, and trends. Urban Aff. Rev. 2003, 38, 325-355. https://doi.org/10.1177/1078087402238805.
  2.  Fallah B. N.; Partridge M. D.; Olfert M. R. Urban sprawl and productivity: Evidence from US metropolitan areas. Pap. Reg. Sci.2011, 90, 451-472. https://doi.org/10.1111/j.1435-5957.2010.00330.x.
  3.  Qin M.; Liu X. Y.; Li S. L. China's "Mystery of Urban sprawl"——Spatial Panel Data Analysis from the Perspective of Government Behavior.Economic Perspectives2016, 7, 21-33, (in Chinese).

Furthermore, we have polished and consulted native English speakers for paper revision before the submission this time. We obtained assistance from MDPI English Editing Services for the revision of English grammar, spelling, sentence structure, and organization quality. This document certifies that the manuscript listed below was edited for proper English language, grammar, punctuation, spelling, and overall style by the highly qualified native English speaking editors at MDPI English Editing Services.

Round 2

Reviewer 1 Report

Dear authors,

thank you for your response to my review and for revising the manuscript accordingly. Most of the remarks in my previous review have been addressed and improved the manuscript significantly from my point of view. The remaining issues have been explained and justified sufficiently.

I have one minor comment before the manuscript is ready for publication in MDPI Land.

The flow chart diagram should appear at the beginning of the Methods section. Possibly also include references to the respective sections of the manuscript in this figure.

Author Response

Dear Reviewers,

Thank you for your letter and your comments on our manuscript entitled Spatio-temporal pattern and driving factors of urban sprawl in China (Manuscript ID: land-1454026). Those comments are all valuable and have been very helpful for revising and improving our paper and providing important guidance for our research. We have studied the comments carefully and have made corrections accordingly, which we hope will be approved. The revised portions are marked in revised mode in the paper. The main corrections to the paper and the responses to the reviewers’ comments are as follows:

Thank you for your response to my review and for revising the manuscript accordingly. Most of the remarks in my previous review have been addressed and improved the manuscript significantly from my point of view. The remaining issues have been explained and justified sufficiently.

I have one minor comment before the manuscript is ready for publication in MDPI Land.

The flow chart diagram should appear at the beginning of the Methods section. Possibly also include references to the respective sections of the manuscript in this figure.

Response: We highly appreciate your positive comments. Special thanks to you for your valuable suggestion. We have modified the relevant section as recommended up to date. We added a simple introduction to the methods section, including the three sections mentioned in the flow chart diagram: the urban sprawl measurements, the spatial and temporal patterns exploration and the driving factors analysis. And, we adjusted the position of the flow chart diagram. The added content is as follows:

  1. Methods

The method involved three steps: (1) quantitative measurement of urban sprawl in China after data processing; (2) exploration on the spatial-temporal patterns of urban sprawl using the ESTDA method; (3) estimation of the impacts of economic, social and government regulation factors on urban sprawl using Geodetector model. For a more intuitive presentation of the workflow, a flow chart diagram is shown in Fig. 1.

Figure 1. Flow process chart.

Reviewer 2 Report

No other comments to make.

The article can be accepted as it is.

Author Response

Dear Reviewers,

Thank you for your letter and your comments on our manuscript entitled Spatio-temporal pattern and driving factors of urban sprawl in China (Manuscript ID: land-1454026). Those comments are all valuable and have been very helpful for revising and improving our paper and providing important guidance for our research.

No other comments to make.

The article can be accepted as it is.

Response: Thanks very much for your kind work and agreeing to publication of our paper. On behalf of my co-author, we would like to express our great appreciation to you.

Reviewer 3 Report

The efforts made by the Authors in order to improve the work are apprecciated. The research is clearer now and can be published.

Author Response

Dear Reviewers,

Thank you for your letter and your comments on our manuscript entitled Spatio-temporal pattern and driving factors of urban sprawl in China (Manuscript ID: land-1454026). Those comments are all valuable and have been very helpful for revising and improving our paper and providing important guidance for our research.

The efforts made by the Authors in order to improve the work are apprecciated. The research is clearer now and can be published.

Response: Thanks very much for your kind work and agreeing to publication of our paper. On behalf of my co-author, we would like to express our great appreciation to you.
